# "*We Are Tired*"—The Sharing of Unpaid Work between Immigrant Women and Men in Portugal

Estefânia Silva [1,2,*], Cláudia Casimiro [2,3], Cristina Pereira Vieira [2,4,5,6], Paulo Manuel Costa [4,6,7,8], Joana Topa [1,2], Sofia Neves [1,2], Janete Borges [1,9] and Mafalda Sousa [10]

1. Department of Social and Behavioral Sciences, University of Maia, 4475-690 Maia, Portugal
2. Interdisciplinary Center for Gender Studies (ISCSP-ULisbon), 1300-663 Lisbon, Portugal
3. Institute of Social and Political Sciences, University of Lisbon (ISCSP-ULisboa), 1300-663 Lisbon, Portugal
4. Department of Social Science and Management, Open University, 4200-055 Porto, Portugal
5. Centre of Global Studies in the Anthropocene, Sustainability and Development, 1669-001 Lisbon, Portugal
6. Centre for the Study of Migration and Intercultural Relations (CEMRI/Uab), 1250-100 Lisbon, Portugal
7. Centre for Functional Ecology—Science for People & the Planet (CFE), 1250-100 Lisbon, Portugal
8. Associate Laboratory TERRA, 1250-100 Lisbon, Portugal
9. School of Health, Polytechnic of Porto, 4200-465 Porto, Portugal
10. Faculty of Psychology and Educational Sciences, University of Porto, 4200-135 Porto, Portugal
* Correspondence: egsilva@umaia.pt

**Abstract:** In this article, we intend to understand and discuss how immigrant men and women living in Portugal perceive their contributions to the performance of unpaid work and how they try to deal with the situation of the greater burden on women. To this end, a qualitative methodology was used to conduct an exploratory study with 10 focus groups of immigrant men and women in five regions of the country: North, Centre, Lisbon, Alentejo and Algarve. The participants, 43 females and 27 males, were aged between 19 and 80 years. From the discourse of the immigrant participants in this study, it could be concluded that the division of unpaid work between immigrant women and men is not equal, as their statements evidenced a greater responsibility and overload on women. From a traditional vision of gender roles, a persistent dichotomy of two worlds could be perceived, based on a "naturalized" vision of the social roles of gender and on a distribution grounded in biological differences. In parallel, discourses show a change in the sharing of household chores and childcare. However, this does not always occur regularly and appears very much associated with the entry of women into the paid labour market.

**Keywords:** immigrant men and women; sharing unpaid work; Portugal

## 1. Introduction

In the last decades, social sciences have been addressing the relationship between unpaid work and gender asymmetries (Kabeer 2016). Unpaid work includes all tasks performed within the scope of the household and which have an economic value, i.e., which can be assigned to third parties for a consideration and are performed by the people in that household without any monetary cost (Cunha and Atalaia 2019). Thus, unpaid work includes both care tasks, in which family members with more limited autonomy, particularly children or the elderly, are accompanied and supported, and domestic tasks, which comprise cleaning and maintenance of the home and day-to-day management, such as shopping, laundry care or meal preparation (Jung and O'Brien 2017).

It is estimated that the monetary value of unpaid work in Portugal in 2019 was somewhere between EUR 39.8 billion and EUR 77.7 billion, depending on whether the reference value is the minimum wage or the average earned value—if this value were integrated into the Portuguese gross domestic product, it would correspond to a growth between 18.6% and 36.2% (Perista and Perista 2022). As concerns unpaid work, it is mostly

performed by women, as a result of a division of tasks between the couple that reproduces traditional gender roles (Pailhé et al. 2021). According to OECD (2016), worldwide, women are the main persons responsible for care work as a primary activity, and when compared to men, spend around twice as much time (or more) on care work.

In the course of the transformations of society in the modern world, particularly since the 1970s, it can be understood that there was a change of perspective and image, with the passage from the notion of nature–woman, conditioned by her biological functions and by a certain destiny, to individual woman, aligned with the idea of feminine agency and empowerment (Torres 2001, 2002). Despite this, it makes sense in terms of social practices and values to continue to perceive that there is a sense of gender constructed as two hierarchized social existences (Bourdieu 1999). Kimmel (2000) also talks about the notion of a *gendered* family, since for this author gender crosses the different dimensions of family life from conjugality and parenthood to the different tasks of housework and care.

Torres (2001, 2002) also points out that, based on definitions of places and roles associated with unpaid work, women experience objectively unfair situations of domestic work overload, as if it were a natural or inevitable fate.

Even if it is argued that the mass entry of women into the labour market would reduce the difference between the number of hours that men and women devote to domestic chores, it is also argued that the use of appliances was a way of freeing up unpaid working time. The latter view is not consensual, since the use of appliances reinforces the maintenance of "gendered" divisions, as their impact on the time spent on household chores tends to benefit men more, as women continue to take on most of the unpaid work (Bittman et al. 2004). For these authors, the negotiation of gender roles explains the change in domestic work more than the support of appliances in the private space. In this context, it can be realized that it is women who continue to spend more hours in the domestic space (Birch et al. 2009). In this space, women continue to maintain a profile of tasks associated with caring, namely for their children and other dependent family members, as well as daily routines, such as meal preparation or laundry (Elson 2017; Matias et al. 2012; Torres 2006). The birth of children tends to foster a more traditional division of gender roles and division of tasks between the couple (Birch et al. 2009; Craig and Mullan 2010; Argyrous et al. 2016).

Although the behaviours of Portuguese society follow the development of modernity, with women playing an increasingly active role outside the home[1] and men being increasing involved in family dynamics and parenting (Perista et al. 2016), the transformations occur slowly, particularly with regard to gender roles (Wall 2005). This author mentions that significant gender asymmetries are present in professional trajectories, given that it is men who always work full-time and rarely interrupt their professional life; while on the contrary, it is women who most present alternate paths of full-time work in their trajectories, with part-time work or with domestic work, i.e., unpaid work that is not compensated by a salary (Alonso-Almeida et al. 2017). Amâncio (1994) refers to the fact that they "naturally assume models of behaviour" that are socially imposed on them and that reinforce the differences between the world of paid and unpaid work. There are studies that show that, in Portugal, it is women who continue to take on most of the unpaid work (Amâncio 2007; Perista et al. 2016; Torres et al. 2013).

A recent study carried out in Portugal by Perista et al. (2016) with a sample of 10,146 people surveyed shows that the total amount of daily unpaid work done by women is, on average, greater than the time spent by men. According to the study data, women dedicate an average of 4 h and 23 min per day, while men dedicate approximately 2 h and 38 min per day.

In other words, women spend 1 h and 45 min more per day than men performing unpaid work, which means that in a week, the time spent by women in the performance of additional unpaid work is 12 h and 25 min (Perista et al. 2016).

According to International Labour Organization (ILO) data from 2019, Portuguese women spend 298 min per day on paid work and 77 min on unpaid care work. Compared to other European countries, Portugal seems to have a better situation, as women dedicate only

21.3% of their time to unpaid care work. In contrast, in France and Spain this percentage is higher, with women devoting about 57% and 38%, respectively, of their time to unpaid care work (Charmes 2019).

Although there is already information about gender inequalities, namely about the unequal way men and women carry out their activities and use their time, we find that these issues are still little explored, specifically when we think about them from the perspective of immigrant people.

Despite the changes observed in the dynamics of immigrant families in recent years, scientific research (e.g., Farré et al. 2023) has been showing that gender disparities still affect women's and men's lives, depending on personal, social, cultural and political factors. Although women had gained more autonomy regarding their decisions, both in family and labour contexts, as they maintain the main responsibility for domestic tasks and childcare, they still bear a disproportionate burden when compared to men (ILO 2019).

In the current organization of contemporary society, immigration is associated with globalization flows and reflects an increasing dynamism, especially in some parts of the world, where the market economy is more fragile and where living conditions are based on increasing poverty. For migrants, access to the labour market assumes a fundamental importance in the conditions of survival, integration and inclusion in receiving countries as well as in the contribution they advocate for their countries of origin (de Ferreira et al. 2017; IOM 2022; Pereira and Esteves 2017). Whether through family reunification or as an autonomous pathway, thousands of immigrants move to other countries in order to improve their quality of life (IOM 2022); it is assumed that immigration will be a way to respond to the limitations they encountered and the possibility of upward mobility (Van Hook and Glick 2020).

In recent decades, Portugal has configured a cartography of hope, making itself a destination country for thousands of migrants who move from the most distinct places (Topa 2023). According to the latest data from the Foreigners and Borders Service in 2022, there were 781,915 citizens with valid residence permits in Portugal, 47.6% being female and 52.4% male (SEF 2023).

In this context, immigrant people, despite being in the physical space of another country, the receiving country, tend to reproduce the gender role division of the country of origin, especially in the initial phase of the migration process (Franck and Hou 2015; Paljevic 2013). Although immigrant people cannot be seen as a homogeneous category, since they come from different countries and cultures, immigrant people tend to reproduce the culture of the country of origin in the host country, so that if they come from more traditional countries, they tend to repeat more traditional values in the host country, even if it is not as traditionalist. Because of this situation, the way people share unpaid work will tend to be more unbalanced, with men participating less, compared to those who come from countries where gender equality is higher (Marcén and Morales 2019; Blau et al. 2020).

However, the way in which the migratory pathway takes place may influence the sharing of tasks and the time dedicated to them, particularly when women manage to access the labour market. The couple tends to lose the social and family support network that they had in their country of origin and that could be used as support, for example, for childcare or for cooking meals (King-Djardin 2019; Liu 2010).

At the same time, the cost of living tends to increase, which implies that the household has to do without certain services, for example, buying meals outside or hiring people to carry out domestic and care tasks (Liu 2010). In this way, and in comparison with the country of origin, the number of hours of unpaid work tends to increase. This implies a greater burden for women, even though it has been found that at the same time, when women also work outside the home, there is a gradual involvement on the part of men (Paljevic 2013). Despite this, the greater participation of men results from a need to adapt and not from a change in the understanding of traditional gender roles and the division of tasks (Paljevic 2013), not least because the relative improvement in the economic situation of immigrant women entering the labour market may not be reflected in the relative reduction

of their work at home, unlike what happens, for example, with Portuguese women (Fendel and Kosyakova 2023). These authors' analysis explains that immigrant women are not able to take advantage of their relative better economic situation and reduce their relative contribution to unpaid work. This is not only for cultural reasons, but also due to issues that have to do with the precariousness of paid work, access to work opportunities with fewer hours of paid activity, the fact that they have less bargaining power in remuneration after taking a lower income and the fact that they may have a larger household size. Aggravated by the professional opportunities of immigrant men and women, which are often restricted to secondary segments of the labour market and labour-intensive sectors (Gundert et al. 2020; Kogan 2011), women, particularly mothers, in general accept paid jobs that allow them to have greater flexibility and shorter working hours, for example (Hakim 2006), a situation that may facilitate a better work–life balance for them. Even so, it remains to be seen how immigrant women and men deal with time management and the reconciliation of paid and unpaid work in Portugal. Therefore, in this article, we intend to understand and discuss how immigrant men and women perceive their different contributions to the performance of unpaid work and how they try to deal with the situation of greater burden on women.

*The Current Research*

The present study is part of the project Boomerang—"Study of the perceptions of the economic impact of unequal sharing of unpaid work in the lives of immigrant women and men in Portugal". Its aim is to characterize the perceptions of the economic impact of unequal sharing of unpaid work and divorce in the lives of immigrant women and men living in Portugal and to analyse their effects on the conciliation of professional, personal and family life.

## 2. Methods

### 2.1. Participants

A total of 70 participants were involved in this study; 43 were female and 27 were male. Forty-three immigrant women (61.4%), aged between 19 and 80 years, participated in the focus groups (M = 43.49; SD = 13.53). Of these, 19 participants were of Brazilian nationality (44.2%), 11 were Cape Verdean (25.6%), 11 women (25.6%) were Ukrainian and 2 were of dual nationality (4.7%). These immigrant women had lived for an average of 14.7 years in Portugal. Regarding marital status, 23 participants were married (53.5%), 4 lived with a partner (9.3%), 6 were single (14%), 8 were divorced (18.6%) and 2 were widows (4.7%). In terms of the level of education, 6 women had primary education (14%), 12 had secondary education (27.9%), 17 had a university degree (39.5%) and 8 had a master's degree (18.6%). Immigrant women had an average of 1.7 children, with the average household being composed of 3 persons. Finally, in relation to their legal situation of residence in Portugal, 40 women were in a regular situation (93%) and 3 were in an irregular situation (7%).

Regarding the men, 27 participated in the study (38.6%) and were aged between 27 and 69 years (M = 44.59; SD = 13.34). Seventeen were of Brazilian nationality (63%), two were Cape Verdean (7.4%), two were Ukrainian (7.4%) and six were of dual nationality (22.2%). These immigrant men had lived for an average of 12.3 years in Portugal. Regarding marital status, 17 participants were married (63%), 2 lived with a partner (7.4%), 4 were single (14.8%) and 4 were divorced (14.8%). In relation to the level of education, four participants had primary education (14.8%), six had secondary education (22.2%), nine had a degree (33.3%), seven had a master's degree (25.9%) and one had a PhD (3.7%). Immigrant men had an average of 1.6 children, with the average household size being 2.7 persons. Regarding the legality of their stay in the country, 25 men were regular (92.6%) and 2 were irregular (7.4%).

*2.2. Procedure*

By means of contact with immigrant associations, Portuguese non-governmental social organizations that work with immigrants in the country, participants were selected based on the following criteria: being 18 years of age or older, residing in Portugal for at least 1 year, being in an intimate relationship or having been in one in the past, and having Brazilian, Cape Verdean or Ukrainian nationality. One of the reasons for the latter option was that, at the time of the project submission, these were the three most representative nationalities in Portugal. However, the current context of war in Ukraine, which collided with the period of data collection, had repercussions on the participation of Ukrainian people. The dimension criterion was established with a minimum of 6 and a maximum of 12 participants (Bloor et al. 2002; Hennink 2014).

The request for the focus groups was made through direct contact with the participants by telephone, and at the beginning of each focus group, the study objectives were explained, with anonymity and confidentiality being safeguarded. A written informed consent statement was signed by each participant and the study respected all the ethical standards of scientific research with human beings, the Code of Ethics of the Order of Portuguese Psychologists and the General Data Protection Regulation.

Due to this process and following a qualitative methodology, ten focus groups were conducted in five regions of the country: North (two), Centre (two), Lisbon Metropolitan Area (two), Alentejo (two), and Algarve (two). Six were conducted face-to-face in the association and immigrant organizations context, and four were conducted online, via Zoom. Each discussion group was based on structured interviews, with open-ended questions about: (a) migrations, (b) paid work, (c) unpaid work, (d) work–family balance.

The data collection, carried out between March and November 2022, had an average duration of 1 h 30 min to 2 h.

*2.3. Data Analysis*

With the proper authorization, the focus group sessions were audio recorded, later transcribed, and subjected to the thematic content analysis proposed by Bardin (2009). During the coding and categorization process, we adhered to Bardin's (2009) guidelines, which include the following rules: completeness (we ensured that the entire communication was thoroughly examined, leaving nothing omitted), representativeness (the sample must represent the whole), homogeneity (data must refer to the same theme, be obtained with the same techniques and be collected by similar individuals), pertinence (the selected documents were relevant and aligned with the content and objectives of the investigation) and exclusivity (each element was assigned to only one category, avoiding overlap or duplication).

The corpus of analysis and the coding and categorization processes were carried out to ensure intersubjective consensus on the part of the research team members.

**3. Results and Discussion**

Content analyses resulted in the identification of 4 primary categories and 10 secondary categories within the discourses of focus groups' participants (Cf. Table 1).

We will describe, analyse and discuss each one of the categories in further detail using specific quotations.

**Table 1.** 1st and 2nd order categories.

| Primary Categories | Secondary Categories |
| --- | --- |
| Domestic Spaces and its organization | Division and sharing of tasks |
| Care Tasks | Child care |
| Impacts of unequal sharing | Tiredness |
| | Guilt |
| | Overload |
| Causes (of unequal sharing) | Culture |
| | Education |
| | Paid work |
| | Maternity |

The discourses of immigrant men and women regarding the division and sharing of unpaid work between the couple, with or without children, show that the division of unpaid work is not egalitarian; their statements evidence a greater responsibility for women, who are often presented as those mainly responsible for domestic chores. This difference reflects a traditional view of gender roles assumed by couples. It is also based on a Parsonsian vision of the dichotomization of two worlds based on a "naturalized" vision of the distribution of the differentiating and complementary social gender roles between women and men, that domestic tasks are perceived as something "natural" and "universal", with a distribution based on biological differences. In their discourse, women go on to use the justification of the situation of "help" on the part of the man as a kind of reproduction of the division of gender roles in the country of origin, which confirms what is mentioned by Franck and Hou (2015) and Paljevic (2013). This can be explained by the quotes of these four women participants:

> [...] I don't like either cooking or washing the dishes or doing anything. I used to want to do everything; now I am the one who takes care of cleaning my house [...]. (Sofia/65/CV/S)[2]

> At home I'm the one who cooks, my husband doesn't even know how to cook [...] he works, he knows how to do everything, everything that a man should know how to do. (Clara/42/UK/M)

> He did not know how to cook, er, no, he did not move to do anything, in fact. He was not used to it and I also did not have a vision of division because of the upbringing I had, so we ended up not dividing it. In theory it was divided, but not in practice. [...] besides not doing it, I still complained that it was badly done [...] I have a male brother and I see, here, since I was a child there was always separation... eh in my case what I think is that it comes a lot from the upbringing that the person has from a young age ... eh my brother, my mother didn't let him wash dishes, make the b, she said it was a girl thing and a boy thing [...] (Lara/35/BR/S)

> In my experience, it is like this, in Cape Verde, like in the old days, eh, we already know, the woman takes care of the children and the house and the man has to work and bring the income home, so we always see things this way—the woman has to do everything [...] and the man only does what he does if he helps, wants to help and helps [...]. (Valentina/28/CV/S)

When there is task sharing, women take on those tasks that are more routine and more time-consuming, and that implies a continuous effort, such as cooking or cleaning the house. This division of tasks is not always the result of a negotiation between the two people in the couple, and it is assumed as natural for each of them to carry out certain tasks and that the woman has a greater share of unpaid work. In other words, in the division, the tendency of the existence of supposedly innate and distinct capacities and aptitudes for each one stands out. In one of the discourses of a participant in this study, this idea

is reinforced with the demonstration of a kind of gratitude and compensation from her husband:

> [...] do you know why I feel more like doing all the housework? because my husband gives me everything [...]. (Beatriz/41/CV/M)

The change in the division of tasks occurs mainly when the woman manages to get a paid job, or when she becomes aware of the overload to which she is subjected and the injustice of this situation. As specifically demonstrated by participants Manuela and Cecília when they say:

> [...] I say "you stay at home", he said "but I am not a maid", "and I am not the man of the house either", so we have to divide the tasks because otherwise you have to hire a maid for here. (Manuela/53/BR/M)

> [...] now she is at home, now she does much more, just yesterday she cleaned the cooker, I came home "did you see what I did?", "and now, what do you want?" (laughs) and all the years that I did it, right (laughs). (Cecília/59/CV/NMP)

The paid work performed outside by both of them assumes a kind of awareness of inequality and of the need to conciliate the work performed in the shared space by both of them. From the speeches (Sofia and Alice) it is clear that since they started working outside the home, the couple has started to share the tasks more.

> [...] now a person comes home... I already feel a little tired, but when I come home, the first thing I do is take off my coat, wash my hands and go to the kitchen ah, now I don't do so much but still [...] when he came home on the weekend on a Saturday, he would put everything on and I would clean, make the meals and put the clothes to dry, I still do it today, a little but [...]. (Sofia/65/CV/S)

> Today, for example, he cleans the whole house, he just does not like to wash the bathroom, then it is for me, OK, but, for example, we agree on Friday "look, you vacuum the whole house, I clean it", I vacuum, he does not like it and he cleans the whole house, then it is like this. We divide the tasks a lot, he is the one that does the laundry [...] I just do the food and wash the bathroom [...] as I worked all the time outside and he could work in the office, he started to do everything and that became automatic. (Alice/32/BR/M)

In their discourse, however, the need is also manifested for some of these immigrant women to take on jobs that allow them to have a greater flexibility of shorter working hours, which justifies the role of responsibility for domestic chores—a situation that is in line with that stated by Hakim (2006) and Polachek (1981). Even so, in this unequal sharing of tasks, Giovanna's discourses prevail in the idea of attributing to her husband the role more related to the care of the child and less to the tasks of cleaning and tidying the house, which are assumed by the husband himself.

> [...] I have a much more flexible schedule, I can stay much longer at home [...] I always end up staying much more at home, I have more ah!!! ah!!, the responsibilities of our son, I'm usually the one who takes, who brings and who does things and even here inside the house I also end up working more than, than he does. Not because he doesn't want to, or doesn't like to, but effectively because he is not here. [...] he also has less free time, so sometimes I prefer him to spend the time he is at home with, with our son than sometimes doing something around the house. (Giovanna/36/BR/M/)

The birth of children normally leads to a greater participation of men but does not reduce women's unpaid workload. As Valentina mentions, before becoming a mother she did all the housework by herself; or as Marina mentions, when she shows that she becomes aware of the unfairness of leaving all the work to her, when both of them work and now she also has to look after the child; or as Liz mentions, when she mentions that problems

arise particularly when the children are small—showing, comparatively, that there is more work.

> [. . .] before becoming a mother I always did it alone, and I said so 'no, it can't be'. And after I became a mother I had to share, because it is very hard to reconcile work, because I practically only stayed at home for a month, had the baby and immediately went to work, so I said "it's very tiring working and then doing things inside the house with only one person". So I had to adapt to that [. . .] (Valentina/28/CV/S)

> [. . .] I taught him (laughs) I taught him to make food, nowadays he knows how to make rice, pasta, beans, season meat, he knows [. . .] so he saw that now, let's say, he has matured, he sees that he has to do it because if he doesn't do it, it is also unfair to leave it only for me because I am not alone for this either because I have to do other things, especially now with a child, everything is much heavier, many more things to do, much more responsibility. (Marina/33/BR/M)

> [. . .] the children are already raised, I think, I think that a good part of the problems, which is when we have small children [. . .]. (Liz/45/BR/NMP)

The lesser participation of men is sometimes compensated with the involvement of the children in the domestic chores, when they are older, in order to reduce overburdening of the woman. But still, the notion of a gendered family remains, (Kimmel 2000) since in the different dimensions of life the domestic work tasks are given to the females in the private space—in the case when it cannot be taken by the woman it is done by the daughter—as Beatriz and Alana describes:

> [. . .] my husband doesn't have a schedule for coming home—he is a little bit lazy because he doesn't like to make the food (laughs), he likes to make a mess in the kitchen, he doesn't like to make food (laughs), and then, if I work at night, my daughter who is here makes the food. She is sixteen but she does the cooking—she does everything [. . .]. (Beatriz/41/CV/M)

> [. . .] daughters are easier, independent [. . .] mine does, for example, vacuuming the house, washing the floor, she does it. (Alana/40/UK/M)

But change also occurs due to education and the change of mentalities and practical values that permeate family life in the contemporary context, breaking very clearly with the older generations. As one of the Cape Verdean women (Cecíl) mentions in her speech, for example, when she says that men with values from other times have no chance to adapt to the new times. This leads us to understand that, on the contrary, younger men adopt a pattern of behaviour that adapts to more flexible, adjustable and negotiable roles.

> The men of yesteryear have no cure, they have no cure anymore (laughs) [. . .] then my son helps, he makes his wife lunch, makes dinner, tidies up the house; both work and everybody does things at home, everybody collaborates [. . .] how many times I went to his house on a Saturday . . . he was vacuuming the house and tidying it. (Cecília/59/CV/NMP)

However, there are statements that show us the continued devaluation of unpaid work and the devaluation of women's work. In other words, there is a type of devaluation of "feminine time", time for domestic work and unpaid care work which is not adequately valued. This understanding of relative importance ends up making this work invisible, as Mariana illustrates:

> It's that saying, right. . . "I work, but my wife doesn't work", then comes the question "What does your wife do? My wife doesn't work. Only me". . . understand. . . so I have this. . . I already have this from my family. (Marina/33/BR/M)

> [. . .] and he says "but you don't work, I do" and I showed him "now you see, the nanny costs one hour, less than five euros, minimum, yes? Then I work all

day and at night it costs, so you you you you you, cleaning costs you you you you you, ah so wash the dishes ah wash the clothes you you you you you and I earn ah better, no, yes, I earn better, more than you because a woman does many things and a man may not see, like this is clean, this is good […] so this is very important for a man that he can see that you work a lot and that costs, only nobody pays you for that. (Olga/41/UK/M)

The "invisible" overload of work for women has consequences in the present with the manifestation of tiredness.

[…] we are tired. (Cecília/59/CV/NMP)

In the discourse, it is clear that, in addition to tiredness, there is a silent demand from the older women in the family (mother and mother-in-law). For example, one of the Brazilian participants, Mariana, explains that there is a kind of silent demand. The overload with domestic and family work, without recognition of the effort, does not allow much free time and when it exists it is assumed with a feeling of guilt.

[…] I feel that there is a demand […] even if it is silent, but I think there is that, as he works outside, that he works the whole day outside, "then when he gets home he has to have the right to rest, he has to have the right to stay home and do that", I don't know what, and I feel this demand, even if it is silent from my mother, from my mother-in-law, so sometimes I feel obliged to do more because he spends more time outside the house than I do […] it is difficult for me to have my free time and often, sometimes when I want some free time I feel a little guilty (laughs) […]. (Giovanna/36/BR/M)

The men's discourse reveals that they are aware that there is an unequal sharing of unpaid work. In other words, there is a threefold understanding on the part of men: (i) the need for a division with greater investment in participation on their part and whose path is to assume the tasks in an equal manner—this division may not exist in a rigid and prior manner, but it can happen with equal responsibility for space and the family; (ii) the need to support overburdened women with a sense of help—to help without effectively seeking to eliminate the imbalance of the division; (iii) there is no need for change, that is, the "naturally" unbalanced distribution is accepted, legitimised by women's competences. We sought to understand how the participants in the study deal with this complexity. In this sense, we found three different types of assumptions that deserve our attention:

First, there is a group of men who suggest the need to change the terms of sharing unpaid work, with them taking a more active role. This discourse of change towards a more balanced distribution of tasks is associated with a feeling of equal responsibility for the space and for the family, which, for example, causes household chores to be performed by whoever comes home first, as the following discourses show:

[…] if both of them work, both of them take care of the house, if only one works, then the house needs to be taken care of by the one who isn't working. So, not that we debated about this, it was automatic […]. (Rafael/50/BR/M)

[…] and if she starts working, which I believe she will, then we will really have to divide things, the, the, the housework […] then we have to, we have to have a division […]. (Daniel/35/BR/M)

[…] "At home I help my wife doing this, doing that", no, no, no, you are not helping your wife, you are helping yourself, because you are the one that has to do it, first because the house is also yours […]. (David/38/CV/M/B)

Alternatively, as one of the Brazilian men (Matheus) mentions, it is possible that one of the members of the couple may contribute more at a certain moment and the other one at another moment, but without this being associated with a previous and rigid division of tasks between the couple.

> [. . .] [there is a] dynamic balance [. . .] [that] does not] always have to be equal for equal [. . .] it has already created a routine for us, so, ah, I like cooking more, so I'm the one who cooks the meals [. . .] sometimes she cooks breakfast, something for dinner, but normally I'm the one who cooks and I also do the planning [. . .] I make a shopping list and then we go together to the market and do the shopping [. . .] my wife is more in charge of the clothes [. . .] when there are a lot of clothes I also help her to hang, fold, but usually she is the one who takes care of all the clothes. Then at the weekends we usually split up and she is in charge of dusting, sweeping and I am more in charge of cleaning the bathroom, scrubbing the toilet, the shower, the window, everything related to cleaning, and what else? She also takes care of our finances because she has more experience with spreadsheets; I am not very good at that so she also controls the expenses [. . .]. (Matheus/31/BR/M)

Second, there is a group of men who see themselves as someone who "helps". In other words, they view it as "natural" that women have a greater proportion of unpaid work, reducing their role to a kind of supporter who seeks to limit the overburdening of women by helping but without actually seeking to eliminate it. There is a kind of legitimacy here of the imbalance of unpaid work and, in this sense, it is even possible that they consider their support as important and that it should even be recognized. In the words of one of the Brazilian men (Bernardo), this sense of support and recognition of support is implied when he says:

> [. . .] I don't mind helping, even [. . .] the work with cleaning the house [. . .] I clean the bathroom, make the bed, do everything, [. . .] I don't have that difficulty at home anymore [. . .] I always help the wife in whatever I can [. . .] I change a nappy, give a bath, make food, wash dishes. . ." (Bernardo/40/BR/M)

Similarly, two more Brazilian men emphasize help as an important support to be valued, namely when they mention:

> I always try to help when she has to make the food, clean the house. I am very present, but it ends up falling a little more to her, and even to my oldest daughter, who is already 17, right. So, it is always a little more for them. I always stay more behind the scenes, helping, whether I get this, whether I get that, want help here, want help there. . . but it ends up being a little more for them, not exactly for me. (Leonardo/38/BR/M)

> [. . .] I help her, right, but the one who stays at home more is my wife and now she, at the moment, she doesn't work, and she only stays, only takes courses [. . .]. (Daniel/35/BR/M)

Finally, a third group of men suggest that the inequality in sharing unpaid tasks is the responsibility of the woman. This burden on the woman is justified because (i) she has a preference for certain tasks, (ii) she does not let him participate, given that he does not do things the way the woman wants them to be done. Thus, by transferring the responsibility to the woman, these men accept this imbalance and suggest that they can do nothing to change the terms of the sharing of unpaid work. There is, in this perspective, a sense of validated justification and acceptance of a kind of "normalized inaptitude" which means that even though men recognize inequality, they cannot correct it, since it does not depend on them. These ideas are objectively demonstrated in the following statements:

> I try to help, sometimes I get in the way more than help, then: "no, it's too much work for me to teach you, so let me do most of it" and only when she can't do it, then I try to do it. [. . .] but I ask and when she starts to teach me, she says that she has no patience [. . .] "because this is something everybody knows, everybody" [. . .] one of my wife's routines is to do. . . (. . .) the house chores are more her responsibility because of my complete incompetence (. . .) in housework! (Gabriel/60/BR/S)

[. . .] she is the one who does the cooking, even for the sake of survival, if I do the cooking, we will all die (laughs) [. . .]. (Isaac/55/BR/M)

This transfer of responsibility from oneself to the partner also occurs, for example, when sharing is justified by the man's increased tiredness, either due to his work schedule or the completion of professional training, which limits his availability to carry out household chores, as one of the Brazilian men (Leonardo) mentions. Despite this, we would like to reinforce that, at the same time, this discourse shows that there is awareness that the justification of "men's tiredness" may not be sufficient to justify unequal sharing and this finding could be understood as an element of change.

[. . .] Just the fact that the man comes home from work and the woman puts the food on, takes the plate away, makes the food, sweeps, just the woman, that's not, I don't agree. . . you know. . .. I don't agree because often she works outside, she still has to work inside the house, no. [. . .] later I start to understand, but in the beginning, she always let it happen. She always let it happen "oh, don't stay there, you're tired", "let me make the food", "let me dress you" [. . .]. (Leonardo/38/BR/M)

In comparison with the division of tasks in the country of origin, some changes are pointed out which seek to respond to the differences in the couple's lives. This translates into a greater burden for the members of the couple, who have to carry out all the unpaid work, and into a redistribution of tasks between them. Thus, for example, the fact that the couple began to have unequal work schedules forced them to redefine this sharing, just as the fact that the woman was unemployed meant that the responsibility for domestic chores was transferred to her. In addition to all these difficulties, they still have to adapt to change—a situation that, as Luna says, causes "psychological exhaustion".

[. . .] back in Brazil [. . .] we had a, a quiet life like that; we could conciliate the work more easily, the leisure, the tasks, it was easier to conciliate the work with a course, it was easier to take more time for leisure, due to the fact that the jobs were also effectively eh on the same timetable, in the same working hours. But here it is still difficult, so eh we still have exhaustion due to everything being new, a lot of changes, [. . .] so there is a psychological exhaustion eh of also being, being away from everything. (Luna/27/BR/M)

In addition, couples lost the family network they had in their country of origin, while the cost of living determined a new lifestyle and, in this case, the economic impossibility of hiring third parties to perform domestic chores, as these services are more expensive.

[. . .] my parents didn't let us do things, we only worked, my parents were the ones who did things, the food, the clothes, these things, you understand? Everything at home was done by my parents. Eh, it was what they wanted, what they wanted to do, you know? And we only worked. (Daniel/35/BR/M)

Although the couple maintained the practices they had in their country of origin when they moved to Portugal, the contact with another cultural and social situation also had the effect of questioning the behaviours acquired in their country of origin. As can be seen objectively in Sara's speech, when she says that:

[. . .] Cape Verdean men think that eh, just because they work outside, the woman is the housewife (laughs), even though the woman also works, yes, we have this thing. We don't have that culture of sharing the tasks like you do here. [. . .] I think it is, that it was correct to divide the tasks, from taking care of, the children, the children and everything. The correct thing was to really divide and equality, it was equality in everything, right? But we know that because of the culture of each country it is, it is different, right [. . .]. (Sara/40/CV/D)

The same sense is expressed in the discourse of one of the male participants (Rafael, who has lived in the USA for 13 years) who calls behaviours that reject the sharing of

domestic work "machismo". Clearly, this participant refers to cultural change as the justification for changing behaviours and for greater help in sharing daily tasks, namely when he says:

> [...] in the past, for example, my mother-in-law doesn't know how to read and write, because she believed that women are at the cooker; women can't learn to read and write, because what will she read and write? That she will work in the kitchen, she will clean all day, what does she need to read and write for? [...] so as I lived my life in the United States after the age of 13, so I can be like that for having gone there, because there is this, since always [...]. [...] the macho side in Brazil is still very strong today, so, like, they believe in this business that the woman has to come home from work and go make the food and clean the kitchen and "la la la", I think this is more the fault of the mothers than the man himself. Is it machismo?! Yes, but he sees it as a child, he was trained this way, the same way they train the girls to take care of the house [...]. (Rafael/50/BR/M)

## 4. Conclusions

The present research aimed to understand and discuss how immigrant men and women living in Portugal perceive their contributions to the performance of unpaid work and how they try to deal with the situation of the greater burden on women. In concordance with other studies, it was possible to conclude that gender asymmetries are still characterising the division of work between immigrant men and women, with women doing more unpaid work than men.

One of the ideas highlighted in this study is that immigrant men and women do not have an equally divided share of unpaid work, emphasizing the greater responsibility of women, who are often presented as being mainly responsible for domestic chores. Despite this, this inequality is not always assumed, often resulting in ambivalent discourse that basically tries to reconcile the reality of family life in the context of contemporary Western society. This is very much determined by the more active role which women play in the paid labour market, with the persistence of a certain continuity of practices in which gender roles are still based on more traditional assumptions and values (Frank and Hou 2016; Brieger and Gielnik 2021). The best expression of this traditional view is the idea revealed by several men that they "help" the woman in domestic and caregiving tasks, clearly admitting that their participation is not equal—it appears as an optional and irregular assisting procedure—reproducing the gender role division of the country of origin (Franck and Hou 2015; Lightman and Link 2021; Paljevic 2013).

This privileged position of men is also expressed in the justifications they give when they collaborate in domestic and care tasks, such as cooking, for example, in which this is associated with "know-how", in contrast to the woman who does not know how to cook; or else they reveal preferences, which allows men to do the tasks they like. When women have a paid job, they seem to feel this inequality more intensely, given that they have less time available and greater fatigue. However, several interventions suggest that men still do not always have a correct perception of the impact that this inequality has on women, which shows the naturalization and acceptance of these traditional roles. Thus, when a change in the allocation of tasks occurs, it is mainly the result of an "external" factor, a paid job, which forces them to make adjustments in family life.

As such, know-how, preference and time availability allow a traditional vision of "gendered" family life to be justified and maintained, revealed in the unequal distribution of unpaid work between the couple. These arguments sustain and even reinforce gender stereotypes and a certain understanding based on assumptions of the "infantilization" and "menorization" of men, who cannot take on certain roles for which they do not have distinct and unique skills within the couple or can only do what they like. This situation reinforces the idea that family life practices are clearly influenced by different social norms and expectations of gender roles.

Although women continue to assume most of the unpaid work in Portugal (Amâncio 2007; Perista et al. 2016; Torres et al. 2013), the situation of immigrant people, particularly women, presents some specificities by comparison: they often lack a family support network, their financial resources are lower due to the type of tasks and contracts they have, which prevents the external contracting of some domestic and care tasks, and social responses for childcare remain limited.

As potentially more vulnerable, immigrants should benefit from particular measures that could enhance family–work balance. Despite women being those who suffer most of the negative effects of family–work imbalance, measures must be addressed to both genders promoting men's participation in domestic and family issues. Stimulating supportive work environments that offer flexible working hours, paid parental leave, and other family-oriented advantages can permit both men and women to balance work and family responsibilities better. Thus, there is a need for greater public investment in policies that foster equal participation and accountability, in order to contribute to the strengthening of gender equality and the promotion of the well-being of family life, in general, and immigrant women in particular.

Further research on the impacts of unequal work division in immigrant women's health could be an important contribution to prevent the negative effects of the burden they are exposed to.

**Author Contributions:** E.S., C.P.V., P.M.C., S.N., J.B. and J.T. contributed to the conceptualization, writing—original draft preparation and methodology. C.C., P.M.C., C.P.V. and M.S. contributed to the software and formal analysis. All authors have read and agreed to the published version of the manuscript.

**Funding:** Project funded under the EEA Financial Mechanism 2014-2021.

**Institutional Review Board Statement:** The procedure complied with the Code of Ethics and Deontology of the Portuguese Psychologists Association (Código de Ética e Deontologia da Ordem dos Psicólogos Portugueses), the American Psychology Association (APA). The General Regulation on Data Protection (Regulamento Geral de Proteção de Dados) from the European Union was followed. It was not mandatory to submit it to the University Ethics Commission.

**Informed Consent Statement:** Informed consent was obtained from all subjects involved in the study.

**Data Availability Statement:** Not applicable.

**Conflicts of Interest:** The authors declare no conflict of interest.

## Notes

[1]     In Portugal, data show that the employment rate for women is 52.6% (PORDATA 2022).

[2]     Legend used in the identification: name (fictitious)/age/nationality: BR (Brazil); CV (Cape Verde); UK (Ukraine)/marital status: S (single); M (married); NMP (union in fact); D (Divorced).

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
