# Peer review of "We Are Tired”—The Sharing of Unpaid Work between Immigrant Women and Men in Portugal"

_socsci, doi:10.3390/socsci12080460_

Round 1

Reviewer 1 Report

The research topic is very interesting as few studies were conducted on the gender roles in immigrants' families in Portugal. The paper is well-written and clearly-structured. However, the author(s) can improve the manuscript by:

(1) Conduct a more extensive review of literature on the changes in gender roles in immigrants' families 

(2) Explain the rationale for the selection of research participants with Brazilian, Cape Verdean or Ukrainian nationality

(3) Further clarify how gender roles in the home country and the host country might impact the gender roles in post-migration families.

Author Response

Dear Reviewer,

Thank you for giving us the opportunity to submit a revised draft of the manuscript “We are tired” – The sharing of unpaid work between immigrant women and men in Portugal” for publication in the Social Sciences Journal.

I attached the responses to the reviewers' comments and concerns.

Best regards

Reviewer 2 Report

I enjoyed reading this paper and think it can make a nice contribution to the literature if it is revised. Since immigration is only going to continue to increase, the more we understand various immigrant experiences, the better. This paper links up immigrant experiences with the historical discussion of women’s unpaid domestic labor, which can help illuminate various experiences in both areas.

Some terms need to be explained/defined. For example, in the introduction, the authors refer to ‘nature-woman’ as compared to ‘individual-woman’ but do not define the latter.

It might be interesting to know if the value of unpaid work in Portugal is comparable to other European countries (Spain, France, Italy?). Is it about the same, greater, lower?

Page 3, what is meant by “the most distinct world geographies”?

In the Methods section, it isn’t clear why the researchers sought Brazilian, Cape Verdean and Ukrainian nationals for the project. Please explain. In addition, there doesn’t appear to be any data used from the Ukrainian participants, so this needs to be explained.

The focus groups are not described. Where, when, how were they completed, etc. Were interviews done in addition to focus groups, or are the authors conflating the focus group data with the interview data? Were the 70 participants part of the focus groups or were 70 individual interviews done? If all of the data is from focus groups, then I would not say that interviews were done. Unclear.

Later in the paper there is no breakdown of any sort of correlation between the ethnicity of the participants and their experience/thoughts about unpaid domestic labor. Can the authors say anything about Brazilian (or Cape Verdean) immigrants in Portugal and domestic labor? Or is there any correlation between length of time spent in Portugal and beliefs around domestic labor? And perhaps more interesting, is there any argument that can be made about level of education and beliefs about/practice around unpaid domestic labor among these immigrants? I ask because it is argued that people with higher levels of education will more likely try to share domestic labor (perhaps because higher educated women will be working outside the home).

The participants appear to have lived for quite a while in Portugal. Do the authors know if those who have immigrated more recently tend to have a more egalitarian feeling toward domestic labor? Is this a generational shift even among those arriving from Brazil or Cape Verde?

I know these ideas above are touched on on page 7, (“but the change that occurs…”) but might be brought to light more throughout. Also on page 7, “in one of the speeches” – ‘speeches’ is not the right word. “Interviews”? “Focus group discussions”?

I didn’t feel that the reference to “tiredness” on page 8 was developed enough, in particular because it is the title of the paper which suggests that it is a concept of great importance. If this silent demand from older female family members is what is maintaining the inequitable practices of domestic labor and is resulting in widely expressed feelings of fatigue among immigrants, it needs to be more clearly explained and nuanced with data. In other words, if men’s and women’s tiredness affecting the feelings and practices of unpaid domestic labor is the main finding/point of the article, it needs to be set up more from the beginning.

Page 8, “complex and non-linear understanding…” I don’t understand this. What do you mean by how participants deal with “this situation”? Which situation? Be clear. In the last paragraph, it would be helpful if the authors begin with, “FIRST, there is a group of men…” since they are going to review 3 types of assumptions and later refer to the ‘second’ and ‘third’.

I think the paper would be better if the authors set up some vignettes a little more with regard to some of the participant quotes. Perhaps this is not possible (again, not sure if these were interviews or focus groups only). For example, explaining that ‘Marcos, who received his Master’s degree in engineering 6 years before immigrating to Portugal but has since been unable to advance from his work on the floor in a small-scale auto parts factory, said that, “xxxxxxxx”’. The authors could also make their arguments clearer, as suggested above. Let the readers know, for example, if, “Most female immigrants from Cape Verde experienced very little support from their partners. As Beatrice said, “xxxxx”’

Page 12, unclear what the “’minority’ of men” means.

The last paragraph contains the only mention of policy problems or hopes for change. If there is a policy angle to the authors’ findings, that needs to be made clear earlier. In addition, they need to explain what the current policies are that they think should change and why.

Author Response

(The authors gave the same response as above.)
